# Research on the Classification of ECG and PCG Signals Based on BiLSTM-GoogLeNet-DS

Jinghui Li [1,2] , Li Ke [1,]*, Qiang Du [1], Xiaodi Ding [1] and Xiangmin Chen [1]

1  School of Electrical Engineering, Shenyang University of Technology, Shenyang 110870, China
2  College of Telecommunication and Electronic Engineering, Qiqihar University, Qiqihar 161006, China
*  Correspondence: keli@sut.edu.cn; Tel.: +86-024-2549-9250

**Abstract:** Because a cardiac function signal cannot reflect cardiac health in all directions, we propose a classification method using ECG and PCG signals based on BiLSTM-GoogLeNet-DS. The electrocardiogram (ECG) and phonocardiogram (PCG) signals used as research objects were collected synchronously. Firstly, the obtained ECG and PCG signals were filtered, and then the ECG and PCG signals were fused and classified by using a bi-directional long short-term memory network (BiLSTM). After that, the time-frequency processing was performed on the filtered ECG and PCG signals to obtain the time-frequency diagram of each signal; the one-dimensional signal was changed into a two-dimensional image signal, and the size of each image signal was adjusted to input the improved GoogLeNet network for classification. Then we obtained the two-channel classification results. The three-channel classification results, combined with the classification results of the above BiLSTM network, were finally obtained. The classification results of these three channels were used to make a decision via the fusion strategy of the improved D-S theory. Finally, we obtained the classification results. Taking 70% of the signals in the database as training data and 30% as test data, the obtained classification accuracy was 96.13%, the sensitivity was 98.48%, the specificity was 90.8%, and the F1 score was 97.24%. From the experimental results, the method proposed in this paper obtained high classification accuracy, and the classification effect was better than a cardiac function signal, which makes up for the low accuracy of the cardiac function signal for judging cardiac disease.

**Keywords:** ECG and PCG signals; bi-directional long short-term memory network (BiLSTM); GoogLeNet; D-S theory

## 1. Introduction

The existing cardiovascular disease diagnosis and treatment technology means are singular, and the diagnosis and treatment processes are complex, which inconvenience a patient's disease examination. Clinically, a doctor's diagnosis about cardiovascular disease is mostly based on multi-modal detection data, such as PCG, ECG, cardiac color Doppler ultrasound, and so on. ECG and PCG signals play very important roles in the early prediction and diagnosis of heart disease. Although artificial auscultation is a convenient and low-cost cardiac diagnosis technology, doctors need to have very rich clinical experiences. It is easy to make mistakes in diagnoses according to the subjective judgments of doctors, especially for some complex heart diseases. An ECG examination is a common method used in modern cardiovascular diagnosis; when the patient has symptoms such as angina pectoris or myocardial infarction, the ECG signals will change significantly [1]. However, the examination of some diseases will still be less sensitive. When the patient asks the doctor for help, the doctor often makes a preliminary judgment about the heart disease according to the patient's condition. For more serious heart diseases, other instruments need to be checked. This process may lead to the development of the disease, with the loss of the best opportunity for treatment. Moreover, in order to fully understand the health status of the patient's heart, currently, only the detection method of an insertable catheter

can be used in clinical practice, but this method is traumatic and risky to the patient, and requires the hospital to have high technical equipment.

Nowadays, most researchers carry out signal analyses and algorithm development for cardiac function signals, such as an ECG or PCG signal, while there are few related studies based on ECG and PCG signals. Although researchers have fused PCG, ECG, and cardiac impedance signals, most of them treat one kind of signal as the reference of another kind of signal. For example, a cardiac impedance signal is processed with the help of an ECG signal, the PCG signal extracts the feature points with the help of the ECG signal, or simply fuses the ECG and PCG signals to give the heart disease diagnosis result by only a cardiac function signal. For example, Rodion Stepanov et al. determined cardiac hemodynamic parameters by using the combined features of ECG, PCG, and cardiac impedance signals. The relevant parameters of cardiac impedance were obtained with the help of ECG and PCG signals [2]. Jan Nedoma et al. compared heart rate monitoring results by synchronizing the acquisition of the ballistic cardiography, electrocardiogram, and phonocardiogram signals [3]. Mónica Martin et al. improved heart disease diagnosis accuracy by acquiring ECG and PCG signals synchronously [4]. Han Li detected coronary artery disease by a dual-input neural network using electrocardiogram and phonocardiogram signals [5]. Abdelakram Hafid et al. synchronously acquired ECG and cardiac impedance signals with the least electrodes. The parameter calculation of a cardiac function signal was completed by means of an ECG signal [6]. Chakir Fatima et al. used different classifiers to classify the signals by the synchronously collected PCG and ECG signals and compared the classification results with the PCG signals [7]. Huan Zhang proposed the detection method of coronary artery disease, which made use of multi-modal features. The multi-modal feature models were proposed using ECG and PCG signals. Finally, the classification accuracies of these modes were compared [8].

Through the above-mentioned research, it can be found that people have gradually shifted the focus from cardiac function signals to more model cardiac function signals. The simultaneous research on multiple decision-making methods shows the advantages of multi-modal signal research and makes up for the flaw of single model signal diagnosis. Therefore, this paper integrates various technologies to realize the judgments of cardiac health statuses via ECG and PCG signals. This paper proposes the classification method of ECG and PCG signals based on BiLSTM-GoogLeNet-DS (combined with the fusion technology for deep learning). This method can extract the features of long-distance dependent information based on BiLSTM, which can improve the training time and learn the features better, and can produce higher accuracy for long-span time series data. The GoogLeNet model has also achieved excellent results in the field of image recognition with its multi-layer convolutional neural network structure and inception structure. This paper, combined with the advantages of these two deep learning networks, uses the BiLSTM network and GoogLeNet network to extract and classify the features of ECG and PCG signals. The final classification results were obtained by using the fusion technology twice with the help of the improved D-S theory. The experiments show that the proposed algorithm improves the classification accuracy of ECG and PCG signals effectively, and is more accurate than single-modal cardiac function signals.

The structure of this paper is organized as follows: the methodology of the proposed algorithm is presented in Section 2. Section 3 presents the related theory about the BiLSTM network and GoogLeNet model. Section 4 presents the implementation steps of the proposed algorithm. Section 5 presents the experiments and results. The experimental dataset and the parameter settings of the network model are presents. The comparative experiments are set. The evaluation criteria of the classification accuracy, sensitivity, specificity, precision, and F1 score are given. The comparison results with other algorithms are provided. Finally, the conclusions are drawn in Section 6.

## 2. Methodology

This paper mainly selects the PCG and ECG signals collected synchronously as the research objects. PhysioNet published a cardiovascular disease dataset containing both PCG and ECG signals in 2016 [9]. Based on this dataset, this paper proposes the classification method based on BiLSTM-GoogLeNet-DS to predict and classify the signals in the database. Firstly, the ECG and PCG signals are preprocessed, respectively, mainly for filtering. Then the processed signals are sent to the BiLSTM network to perform network fusion and classify the signals. After that, the filtered PCG and ECG signals are, respectively, subjected to wavelet transform to obtain the time-frequency diagram of each signal. Each time-frequency diagram is resized and converted into the image size suitable for GoogLeNet network training, and then classified through the improved GoogLeNet network to obtain the classification results of PCG and ECG signals, respectively. The three-channel classification results, combined with the classification results of the above BiLSTM network, are obtained. The classification decision is made by the fusion strategy of the improved D-S theory. Calculating the probability values of ECG and PCG signals belonging to different categories can lead to the basic probability assignment (BPA) function. The local recognition credibility of each feature is estimated by using the confusion matrix. The weighted correction is carried out when constructing the BPA of the classifier. Finally, the classification results are obtained. The following is the specific implementation process, as shown in Figure 1.

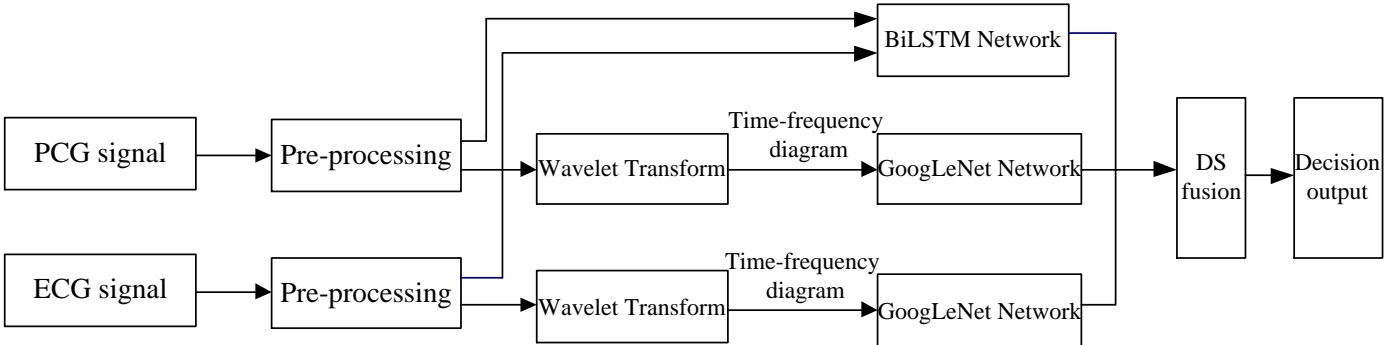

**Figure 1.** Classification method and implementation process of ECG and PCG signals.

## 3. Related Theory

### 3.1. BiLSTM

LSTM (long short-term memory network), which has the advantage of memorability, is an improved model of the recurrent neural network. It can efficiently learn the nonlinear characteristics of the time series. In the internal structure of LSTM, the activation method of the nodes in the hidden layer is very complex. By selectively remembering, forgetting information with relatively small effects, and remembering useful information with relatively large time intervals, it solves the problem of gradient explosion and gradient disappearance of a long time series.

The memory unit of LSTM consists of three gate structures: input gate, forget gate and output gate. The network structure is shown in Figure 2. These gates ensure that information can be selectively passed, and the state of the transmission unit can be obtained by some linear operations, so as to ensure the invariance of information during transmission. By memorizing the information selectively, it can ensure that most of the sample data are effectively used. Its deep features can be learned to diagnose diseases.

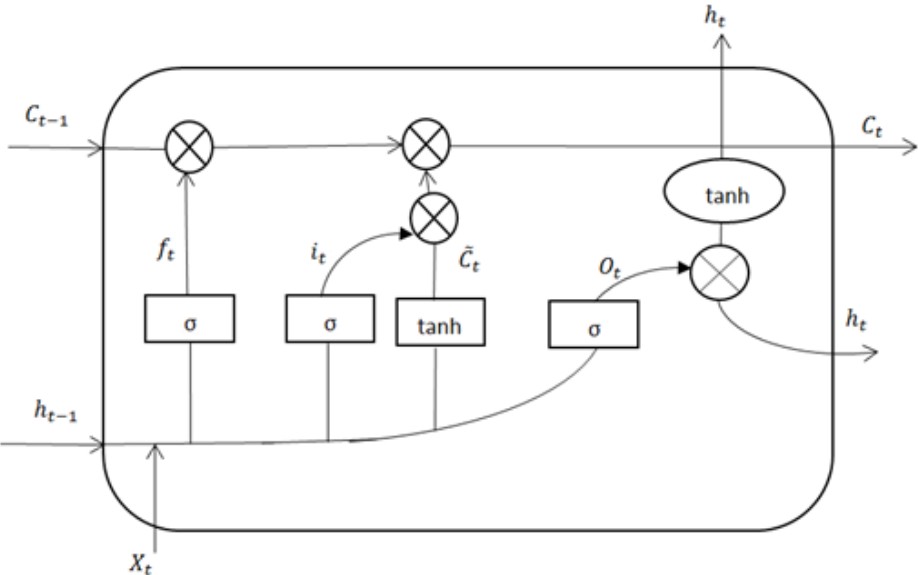

**Figure 2.** Network structure of LSTM.

In LSTM, the sample data will go through the following three stages: the forgetting gate, deciding which information to forget, and which information to retain. Through the input gate, the cell state is updated and the updated information is determined. Through the output gate, the output information is obtained.

The current sample input $x_t$ and the previous sequence output $h_{t-1}$ jointly determine the cell state $c_{t-1}$ of which information needs to be remembered.

$$f_t = \sigma(W_f \cdot (h_{t-1}, x_t) + b_f) \tag{1}$$

In the following formula: $W_f$ is the weight. $b_f$ is offset. $\sigma$ is the sigmoid activation function. $C_{t-1}$ represents the cell state, which is used to save the information of the previous memory moment. $f_t$ represents the forgetting degree of the forgetting gate to the cell state. The input gate updates the cell state and determines the new information to be added to the cell state.

$$i_t = \sigma(W_i \cdot (h_{t-1}, x_t) + b_c) \tag{2}$$

$$\tilde{C}_t = \tanh(W_c \cdot (h_{t-1}, x_t) + b_c) \tag{3}$$

$$C_t = f_t \cdot C_{t-1} + i_t \cdot \tilde{C}_t \tag{4}$$

In the following formula: $i_t$ represents the update degree of the input gate to the new information; $\tilde{C}_t$ is the candidate vector of the current new state information; then the cell state is updated, $f_t \cdot C_{t-1}$ represents the information to be forgotten; $i_t \cdot \tilde{C}_t$ represents the information to be retained; $C_t$ represents the current cell state. In this way, the information is stored in the memory unit, and finally the information is output at the output gate.

$$o_t = \sigma(W_o \cdot (h_{t-1}, x_t) + b_o) \tag{5}$$

$$h_t = o_t \cdot \tanh(C_t) \tag{6}$$

In the following formula: $o_t$ represents the output information of the output gate; $h_t$ is the output of the hidden layer. At the same time, the output $h_t$ will also be fed into the next LSTM unit.

Sigmoid function:

$$\sigma(x) = (1 + e^{-x})^{-1} \tag{7}$$

Tanh function:

$$\tanh(x) = \frac{e^x - e^{-x}}{e^x + e^{-x}} \tag{8}$$

Although the LSTM neural network has a memory function and can learn historical information, it can only learn forward information and cannot make effective use of backward information. The BiLSTM network model is the improvement of LSTM. It can make up for the shortcomings of the above LSTM, and its learning ability has further improved compared to LSTM. The BiLSTM structure is shown in Figure 3. The BiLSTM network has two-direction transmission layers, a forward-transmission layer, and a back-transmission layer. The forward-transmission layer trains the time series forward, the back-transmission layer trains the time series backward, and both the forward-transmission and back-transmission layers are connected to the output layer.

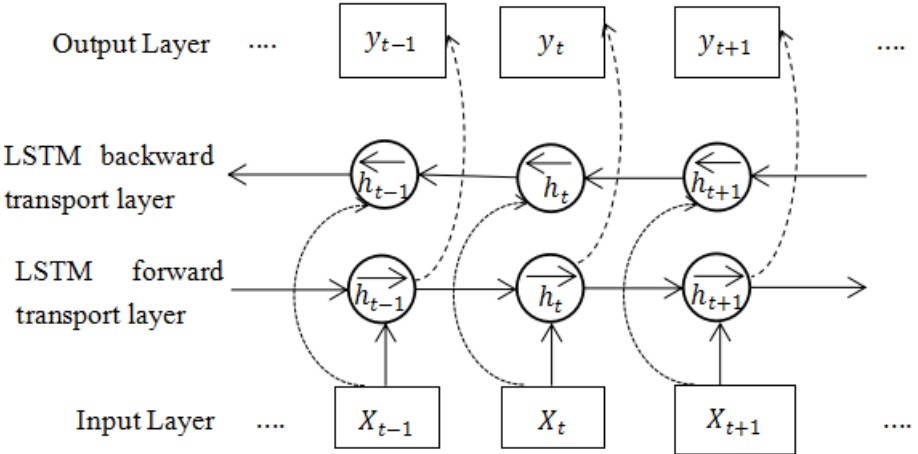

**Figure 3.** BiLSTM structure diagram.

The original signal directly inputs the information to the BiLSTM network layer through the input layer. The input sample signal obtains a value through the forward LSTM calculation and obtains a value through the reverse LSTM calculation. The value transmitted to the hidden layer is determined by these two values.

The formula is as follows:

$$\overrightarrow{h}_t = f(\overrightarrow{w} \cdot x_t + \overrightarrow{v} \cdot \overrightarrow{h}_{t-1} + \overrightarrow{b}) \tag{9}$$

$$\overleftarrow{h}_t = f(\overleftarrow{w} \cdot x_t + \overleftarrow{v} \cdot \overleftarrow{h}_{t-1} + \overleftarrow{b}) \tag{10}$$

$$y_t = g(U[\overrightarrow{h}_t; \overleftarrow{h}_t] + c) \tag{11}$$

In the formula, $\overrightarrow{h}_t$ is the output of the forward LSTM layer, $\overleftarrow{h}_t$ is the output of the backward LSTM layer, the output state of BiLSTM will be spliced and output according to $\overrightarrow{h}_t$ and $\overleftarrow{h}_t$; $y_t$ is the output of the hidden layer after the BiLSTM superposition [10–13].

The original cardiac function signals are directly input into the deep learning network model without manual feature extraction. The deep information is learned through the neural network, and the final classification results are output, so as to realize the classification and diagnosis of cardiac function signals. Since the ECG and PCG signals are time series signals, the relationship between the front and back sequences of signals should be fully considered. The BiLSTM network can take into account the input of forward and backward information. Compared with the LSTM network, the BiLSTM network can make more effective use of the information contained in cardiac function signals. Since the signal values of forward and reverse transmission of different data are different, the BiLSTM

network can use bidirectional information to learn its intrinsic features, so as to improve the classification accuracy.

### 3.2. GoogLeNet Model

Based on the mechanism of convolutional neural networks, many researchers have designed convolutional neural network models for specific problems, such as GoogLeNet, AlexNet, VGGNet, ResNet, and so on. The GoogLeNet model proposed by the Google team is a relatively successful convolutional neural network model. The model has a total of 22 layers of network structure. In the model, in addition to the convolution layer, pooling layer, and full connection layer, there is also the inception structure proposed by the Google team. Through the convolutional kernels of different scales in the inception structure, different image features can be extracted. The GoogLeNet model has achieved excellent results in the field of image recognition with its multi-layer convolutional neural network structure and inception structure.

The core architecture of the GoogLeNet network model has nine inception modules. The initial inception v1 network consists of three convolutional layers of size $1 \times 1$, $3 \times 3$, $5 \times 5$, and a $3 \times 3$ pooling layer (as shown in Figure 4). This structure processes the input images in parallel, and then splices each output result according to the channel, which is used to increase the width of the network and the adaptability of the network to the scale. If the input image has more channels, the convolutional kernel parameters will be more. The computation amount will increase accordingly, so it is necessary to reduce the dimension of the data. Since the $1 \times 1$ convolution does not change the height and width of the image, it can reduce the number of channels. Therefore, $1 \times 1$ convolutional kernels are added before $3 \times 3$, $5 \times 5$ convolutional layers and after the $3 \times 3$ max pooling layer, respectively, to reduce the feature map thickness, as shown in Figure 5.

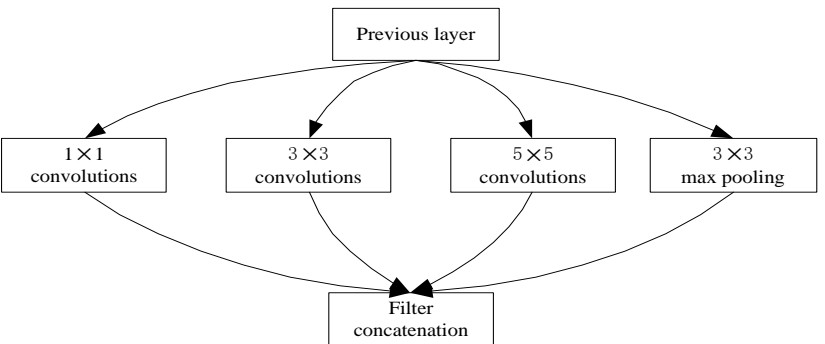

**Figure 4.** Inception network module diagram.

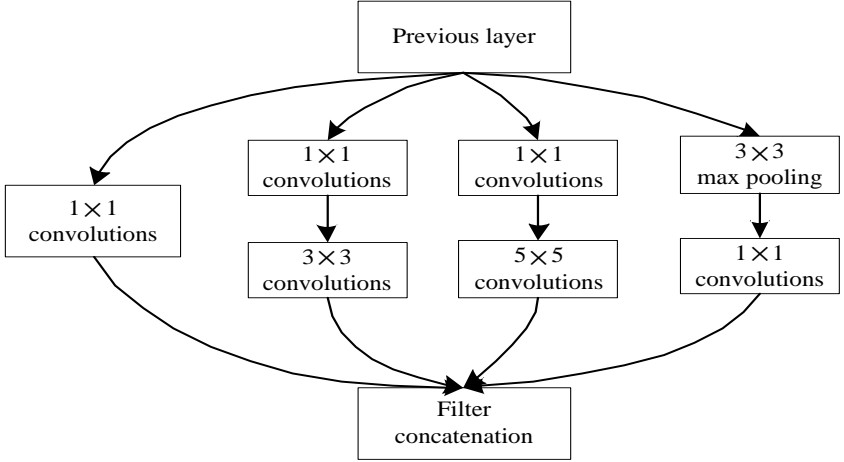

**Figure 5.** Inception network module diagram.

As can be seen from Figure 5, this structure is different from the convolutional neural network of the earlier sequential structure. The difference in the size of the convolutional kernel makes the expressions of the model features more diversified, which greatly improves the ability of the model feature extraction. The GoogLeNet structure is shown in Figure 6. Due to the complex network structure of GoogLeNet, its structure is approximately represented by modules of different colors representing different network functions. The red module represents the pooling layer, the blue module represents the convolution layer and ReLU layer, the green module represents the splicing operation, the yellow module represents the Softmax activation function and the white modules at both ends represent the input and classified output. The red box represents different Inception structures and the green box represents the auxiliary classification structure. From this figure, it can be seen that the GoogLeNet structure contains many similar inception modules. A complex GoogLeNet model is formed by connecting these modules end-to-end [14–18].

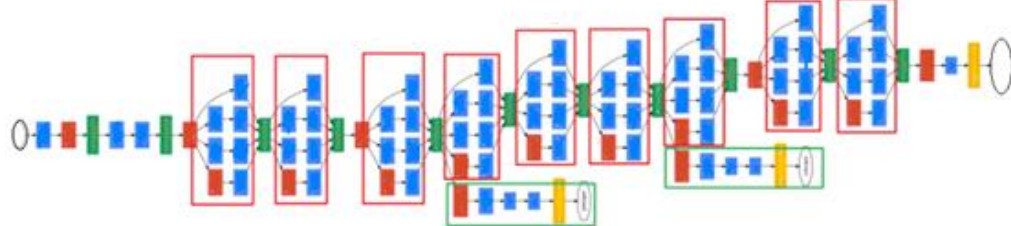

**Figure 6.** GoogLeNet structure.

## 4. Classification of ECG and PCG Signals Based on BiLSTM-GoogLeNet-DS

The ECG and PCG signal classification method based on BiLSTM-GoogLeNet-DS is proposed. Different fusion methods, combined with the network structure characteristics of the BiLSTM network and GoogLeNet network, are used to classify the ECG and PCG signals, which make up for the inaccuracy of disease judgment by using the cardiac function signal. Using the ECG and PCG signal classification, a better classification and diagnosis effect was obtained. The implementation steps of this algorithm are as follows:

Step 1: Obtain the ECG and PCG signals. This paper uses the PCG and ECG signals synchronously collected in the PhysioNet/CINC challenge 2016 heart sound database as the research object. Because the signal acquisition time in the database is different, the length of the signal was unified in the experiment, and 10,000 sampling points of each signal were intercepted for the experiment. The sampling frequency of the signal is 2000 Hz. The database includes 113 normal signals and 292 abnormal signals (405 signals in total). Figure 7 shows the synchronously collected PCG and ECG signals.

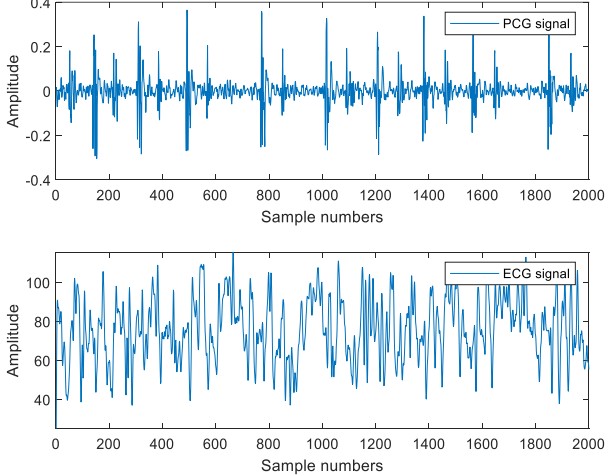

**Figure 7.** PCG and ECG signals.

Step 2: Preprocess the ECG and PCG signals. The main purpose is to filter the signals. The useful information of the ECG signal is concentrated below 20 Hz. If the ECG signal has strong power frequency interference, the signal will generally produce a peak at the power frequency of 50 or 60 Hz in the spectrum. The effective frequency of the PCG signal is concentrated below 300 Hz. In this paper, the ECG signal was mainly subjected to low-pass filtering with the cutoff frequency of 20 Hz; a high-pass filter with the cutoff frequency of 25 Hz and a low-pass filter with the cutoff frequency of 400 Hz were performed on the PCG signal. Since there is too much noise in the ECG signal, the denoising results of the ECG signal are given here. Figure 8 shows the comparison of the ECG signal before and after filtering.

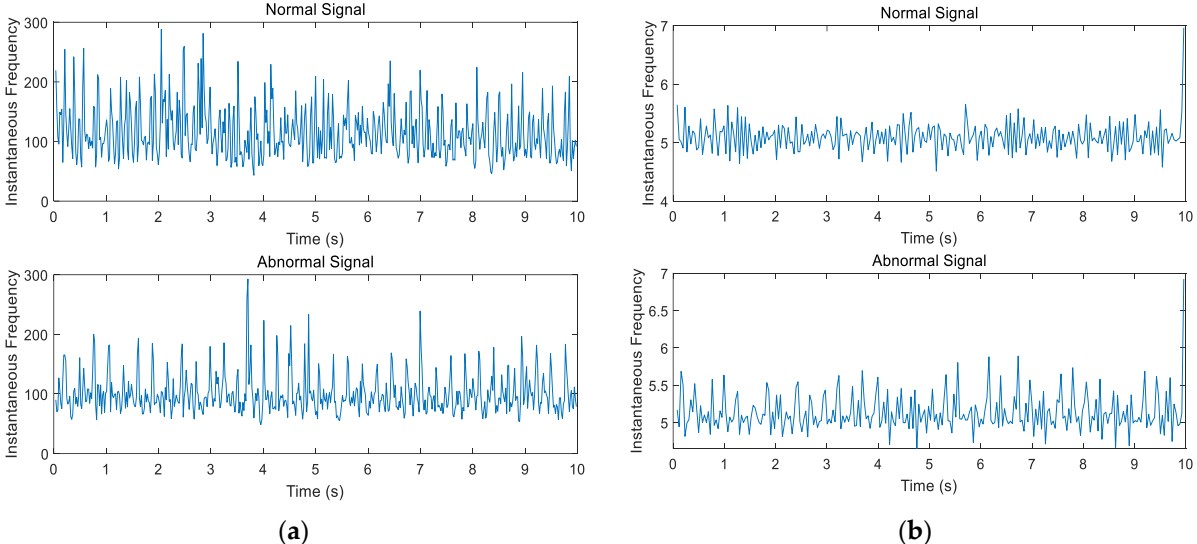

**Figure 8.** Comparison before and after filtering: (**a**) ECG original signal; (**b**) ECG signal after filtering.

Step 3: Use the BiLSTM network for training to create the network model. The filtered signals are trained and classified with a total of 405 samples. Moreover, 70% samples are used for training and 30% samples are used for the test. Then the samples are readjusted according to the labels. The abnormal samples and normal samples are placed separately in a centralized manner and then reintegrated into a new dataset. This setting is convenient for network training and results comparison during the test. The integrated ECG and PCG data are spliced and fused as the input of the BiLSTM network. The optimal network model is obtained through training. Then the classification results are obtained. The BiLSTM network architecture is shown in Figure 9.

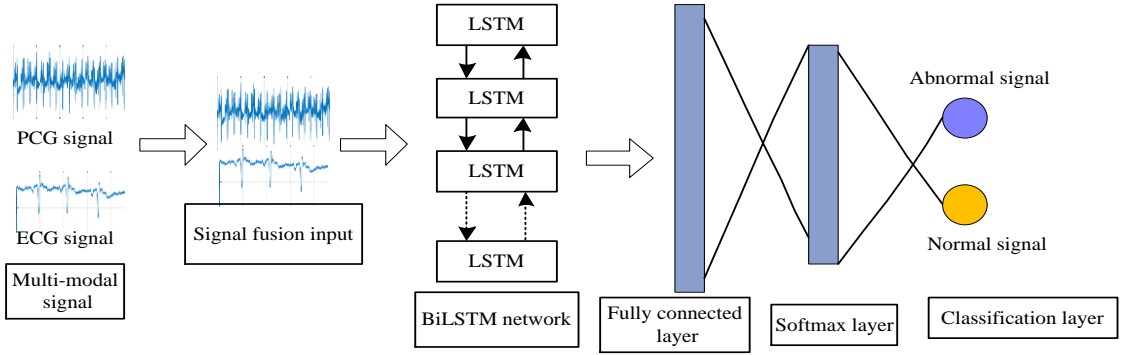

**Figure 9.** BiLSTM network structure.

Step 4: Convert the filtered one-dimensional ECG and PCG signals into a two-dimensional image. The filtered signal is subjected to the continuous wavelet transform to obtain the wavelet coefficients. Then the scale sequence is converted into the frequency sequence. Finally, the wavelet time- frequency diagram of each signal is obtained by combining the time series. The size of the time-frequency diagram is adjusted to 224 × 224, as shown in Figure 10.

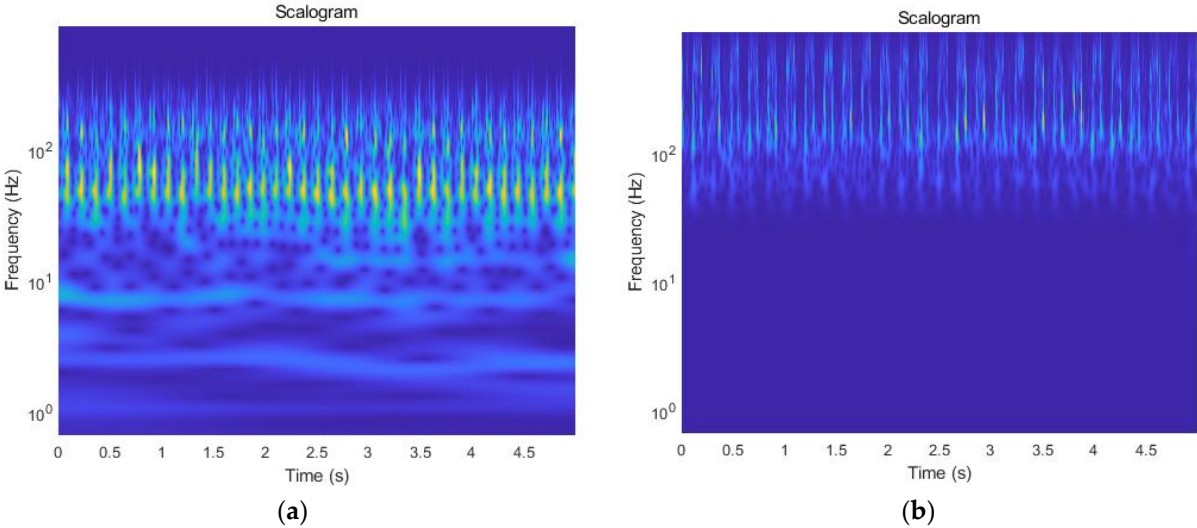

**Figure 10.** Time-frequency diagrams of signals: (**a**) the time-frequency diagram of the ECG signal; (**b**) the time-frequency diagram of the PCG signal.

Step 5: Determine the GoogLeNet network structure and improve its structure. The two-dimensional properties of the time-frequency diagram of each signal obtained above are converted into a three-dimensional form 224 × 224 × 3, which is suitable for the input of the GoogLeNet network. Then 70% of the signals are used for training and 30% of the signals are used for the test. Then determine the GoogLeNet network structure, as shown in Figure 11. The original GoogLeNet network uses different inception modules to generate a total of 144 layers of the network structure. On the basis of retaining the original 140-layer structure, this paper improves the last four-layer structure. Using the new dropout layer, the fully-connected layer, the Softmax layer, and the classification layer replace the original layers. This network structure is used to train the PCG and ECG signals, respectively. The optimal network model is obtained by continuously adjusting the network parameters. Then the classification results of the two-channel signals are obtained.

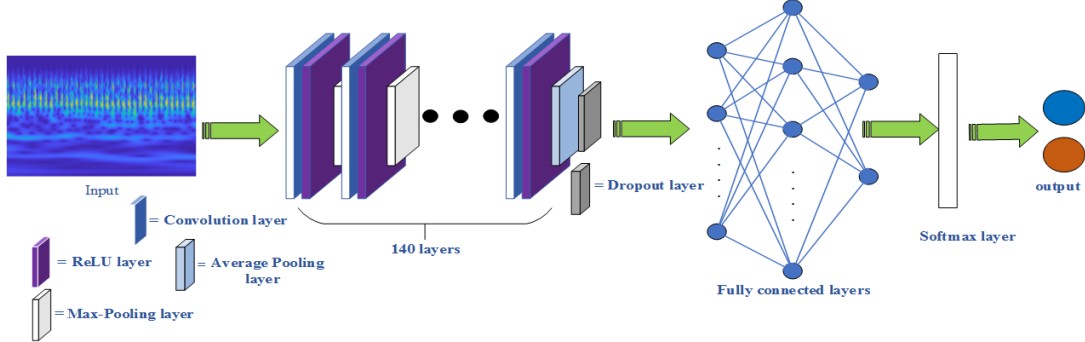

**Figure 11.** GoogLeNet network structure.

Step 6: The three-channel classification results obtained in the above steps are fused at the decision level using the improved D-S theory. According to the obtained classification results, the BPA function under various classification methods is obtained. Then the ECG and PCG signals are fused and determined according to the improved D-S theory. The following is the implementation process.

(1)  Classification framework establishment

The above classification results are used as the evidence of ECG and PCG signal classifications. The classification framework is established, as shown in the formula.

$$\Theta = \{\omega_{normal}, \omega_{abnormal}\} \tag{12}$$

$\omega_{normal}$ represents the normal signal with a classification result, and $\omega_{abnormal}$ represents the abnormal signal with a classification result. The subset of the framework $\Theta$ contains $\{\{\omega_{normal}\}, \{\omega_{abnormal}\}, \{\omega_{normal}, \omega_{abnormal}\}\}$.

Research studies that deposit large datasets in publicly available databases should specify where the data are deposited and provide the relevant accession numbers. If the accession numbers have not yet been obtained at the time of submission, please state that they will be provided during the review. They must be provided prior to publication.

(2)  The BPA function establishment

The normal PCG signal probability value $p(\omega_{1n})$ and the abnormal PCG signal probability value $p(\omega_{1p})$ obtained in Step 5 are used as the BPA function $m_1$ of the PCG signal. The normal ECG signal probability value $p(\omega_{2n})$ and the abnormal ECG signal probability value $p(\omega_{2p})$ are used as the BPA function $m_2$ of the ECG signal. The normal ECG and PCG signal probability value $p(\omega_{3n})$ and the abnormal ECG and PCG signal probability value $p(\omega_{3p})$ obtained in Step 3 are used as the BPA function $m_3$ of the ECG and PCG signals, as shown in the formula.

$$\begin{cases} m_1(\omega_{normal}) = p(\omega_{1n}) \\ m_1(\omega_{abnormal}) = p(\omega_{1p}) \\ m_2(\omega_{normal}) = p(\omega_{2n}) \\ m_2(\omega_{abnormal}) = p(\omega_{2p}) \\ m_3(\omega_{normal}) = p(\omega_{3n}) \\ m_3(\omega_{abnormal}) = p(\omega_{3p}) \\ m_1(\{\omega_{normal}, \omega_{abnormal}\}) = m_2(\{\omega_{normal}, \omega_{abnormal}\}) = m_3(\{\omega_{normal}, \omega_{abnormal}\}) = 0 \end{cases} \tag{13}$$

According to the D-S theory, the probability function values $m(\omega_{normal})$ and $m(\omega_{abnormal})$ of the evidence are obtained, namely, the basic credibility function, as shown in Formula (14).

$$\begin{cases} m(\omega_{normal}) = \frac{m_1(\omega_{normal})m_2(\omega_{normal})m_3(\omega_{normal})}{K} \\ m(\omega_{abnormal}) = \frac{m_1(\omega_{abnormal})m_2(\omega_{abnormal})m_3(\omega_{abnormal})}{K} \end{cases} \tag{14}$$

$K$ is called as the fusion index. The bigger the $K$ value, the higher the support degree belonging to the same category.

$$K = m_1(\omega_{normal})m_2(\omega_{normal})m_3(\omega_{normal}) + m_1(\omega_{abnormal})m_2(\omega_{abnormal})m_3(\omega_{abnormal}) \tag{15}$$

(3)   Implementation of the improved D-S theory algorithm.

The basic principle of the algorithm is to use the method based on local credibility to perform the weighted correction. The classification probability is corrected by the confusion matrix generated previously. Then construct the BPA function of the classifier.

For the classification problem in this paper, it represents the tendency to judge as normal or abnormal. $P\omega(\omega_i|\omega_j)$ represents the support degree of the target belonging to class $\omega_i$. Thus, $P\omega(\omega_i|\omega_j)$ is called the local credibility of the classifier for class $\omega_i$. The BPA function $m(\omega_i)$ about the target belonging to $\omega_i$ is:

$$m(\omega_i) = P_i \times P\omega(\omega_i|\omega_j) \tag{16}$$

$$P\omega(\omega_i|\omega_j) = \frac{\omega_{ij}}{\sum\limits_{j=1}^{k} \omega_{ij}} \tag{17}$$

In the formula, $P_i$ is the probability of belonging to class $\omega_i$ by the previous classification output results. $P\omega(\omega_i|\omega_j)$ is the local credibility information. $\omega_{ij}$ represents the number of samples in which class $\omega_i$ is judged to be class $\omega_j$. k is the category number of signals. The BPA functions are all obtained after the above-mentioned weighted-fusion correction [19–25].

Using the obtained BPA, information fusion is carried out according to Formulas (14) and (15).

Step 7: Obtain the final classification results according to the discriminant criterion by using the previous fusion results.

The discrimination criterion is as follows:

$$\begin{cases} x_i = -1, x_i >= \varepsilon \\ x_i = 1, x_i < \varepsilon \end{cases} \tag{18}$$

where $\varepsilon$ is the set threshold value, $x_i$ is the final classification result, "$-1$" represents normal sample, and "1" represents the abnormal sample.

## 5. Experiments and Results

### 5.1. Dataset

The MIT PCG Database (MTTHSDB) contributes to this study. This dataset comes from the PhysioNet/CINC challenge 2016 heart sound classification competition on the PhysioNet website. PhysioNet acquire datasets (training-a to training-f) from different research institutes. The training-a dataset is provided by MIT and contains 409 samples. Among these 409 samples, 405 samples include ECG and PCG signals, simultaneously. There are 113 normal signals and 292 abnormal signals in the database. The signals are recorded at the sampling rate of 44.1 kHz. They are resampled at 2000 Hz in the post-processing. In the experiment, 70% of the signals in the database were used as the training set, and 30% were used as the test set [26].

The experiment was performed on the computer processor Intel (R) Celeron (R) CPU N3450 1.10 GHz and RAM 4.00 GB. The experimental software was MATLAB 2021a (Natick, MA, USA).

### 5.2. Parameter Settings of the Network Model

1.   Parameter settings of the BiLSTM network model.

The main parameters in the model include the dimension of the input vector, the number of hidden layers, the related parameters of training, and so on. The optimal parameter settings of the BiLSTM network model are shown in Table 1.

2. Parameter settings of the GoogLeNet network model.

The main parameters in the model include the dimension of the input vector, the related parameters of training, and so on. The optimal parameter settings of the GoogLeNet network model are shown in Table 2.

**Table 1.** BiLSTM network model parameters.

| Parameters | Definition | Optimal Value |
|---|---|---|
| sequenceInputLayer | Input vector dimension | 284 |
| Number of hidden units | Number of hidden nodes | 50 |
| Adam | Optimizer | Adam |
| MaxEpochs | Training epochs | 10 |
| MiniBatchSize | Number of samples used per batch | 150 |
| InitialLearnRate | Learning rate | 0.01 |

**Table 2.** GoogLeNet network model parameters.

| Parameters | Definition | Optimal Value |
|---|---|---|
| imgsTrain | Input vector dimension | [224 224 3] |
| dropoutLayer | Random drop rate | 0.6 |
| sgdm | Optimizer | sgdm |
| MaxEpochs | Training epochs | 20 |
| MiniBatchSize | Number of samples used per batch | 15 |
| InitialLearnRate | Learning rate | 0.001 |

*5.3. Evaluation Metrics*

To test the proposed algorithm, the classifications, accuracy, sensitivity, specificity, and F1 score are used to measure the performance. Since the signals are divided into normal and abnormal signals, the abnormal signals are regarded as positive and the normal signals are regarded as negative. If the positive signal is classified correctly, it is regarded as the true positive signal (TP), otherwise it is regarded as the false negative signal (FN); if the negative signal is classified correctly, it is regarded as the true negative signal (TN), otherwise it is regarded as the false positive signal (FP) [27,28].

$$Accuracy = \frac{TP + TN}{TP + FP + FN + TN} \times 100\% \tag{19}$$

$$Sensitivity = Recall = \frac{TP}{TP + FN} \times 100\% \tag{20}$$

$$Specificity = \frac{TN}{TN + FP} \times 100\% \tag{21}$$

$$Precision = \frac{TP}{TP + FP} \times 100\% \tag{22}$$

$$F1\ Score = \frac{2 \times Recall \times Precision}{Recall + Precision} \times 100\% \tag{23}$$

*5.4. Comparative Experimental Setup*

In the experiment, the following classification models are set to evaluate the model in this paper, and the classification performance of the BiLSTM-GoogLeNet-DS model is verified by comparing the models:

(1) BiLSTM model.

The BiLSTM model is composed of two LSTMs in different directions, which can simultaneously capture the front and backward timing information and solve the long-distance dependency problem in the classification process. After the PCG and ECG signals are filtered separately, the signal features are extracted by BiLSTM, and then input into the fully-connected layer and the Softmax layer to realize the classification of the signal. In order to verify the superiority of ECG and PCG signal classification, in addition to inputting PCG and ECG signals into the BiLSTM network model to obtain their respective classification results, the paper also splices and fuses the PCG and ECG signals into the BiLSTM network model for classification. Since the classification effects of ECG and PCG signals are better than that of single-modal cardiac function signals using the PCG or ECG signal, this paper mainly focuses on the classification results of ECG and PCG signals using the BiLSTM model for further processing.

(2) GoogLeNet model.

The GoogLeNet model consists of several inception modules. The GoogLeNet model structure selected in this paper consists of nine layers of inception modules, namely inception-3a, inception-3b, inception-4a, inception-4b, inception-4c, inception-4d, inception-4e, inception-5a and inception-5b. Each inception module includes three convolutional kernels of $1 \times 1$, $3 \times 3$, and $5 \times 5$. When the image data are input into the network, $7 \times 7$ convolution is performed first, then the ReLU layer, pooling layer, regularization, and $3 \times 3$ convolution. After the ReLU layer, pooling layer, and regularization, the data are input into each inception module in turn. The filtered PCG and ECG signals are subjected to continuous wavelet transform to obtain their time-frequency diagrams. The sizes of the diagrams are transformed to be suitable for the input into the GoogLeNet network. The network is used to classify the PCG and ECG signals, respectively.

(3) BiLSTM-GoogLeNet-DS model.

The BiLSTM-GoogLeNet-DS model first splices and fuses ECG and PCG signals into the BiLSTM network to obtain the fused classification results. Then the ECG and PCG signals are, respectively, input into the GoogLeNet network for classification. Based on the above classification results, a total of three channel classification results are obtained. The improved D-S theory is used to fuse the above classification results. Then the classification results are finally determined according to the discriminant criterion.

*5.5. Experimental Results and Analysis*

In order to verify the superiority of the proposed algorithm, the above network models are compared with the BiLSTM-GoogLeNet-DS model, mainly from the evaluation indicators.

This paper improves the last four layers of the GoogLeNet network, and mainly updates the random dropout layer, fully-connected layer, Softmax layer, and classification layer. Using the improved D-S theory, the BPA function is weighted by the method based on confusion matrix. The following are some experimental results. Figure 12 shows the confusion matrix obtained by different models. The confusion matrix is mainly used to weight the BPA function. It solves the problem that the BPA function of the D-S theory is difficult to determine in decision fusion.

Interventionary studies involving animals or humans, and other studies that require ethical approval, must list the authority that provides approval and the corresponding ethical approval code.

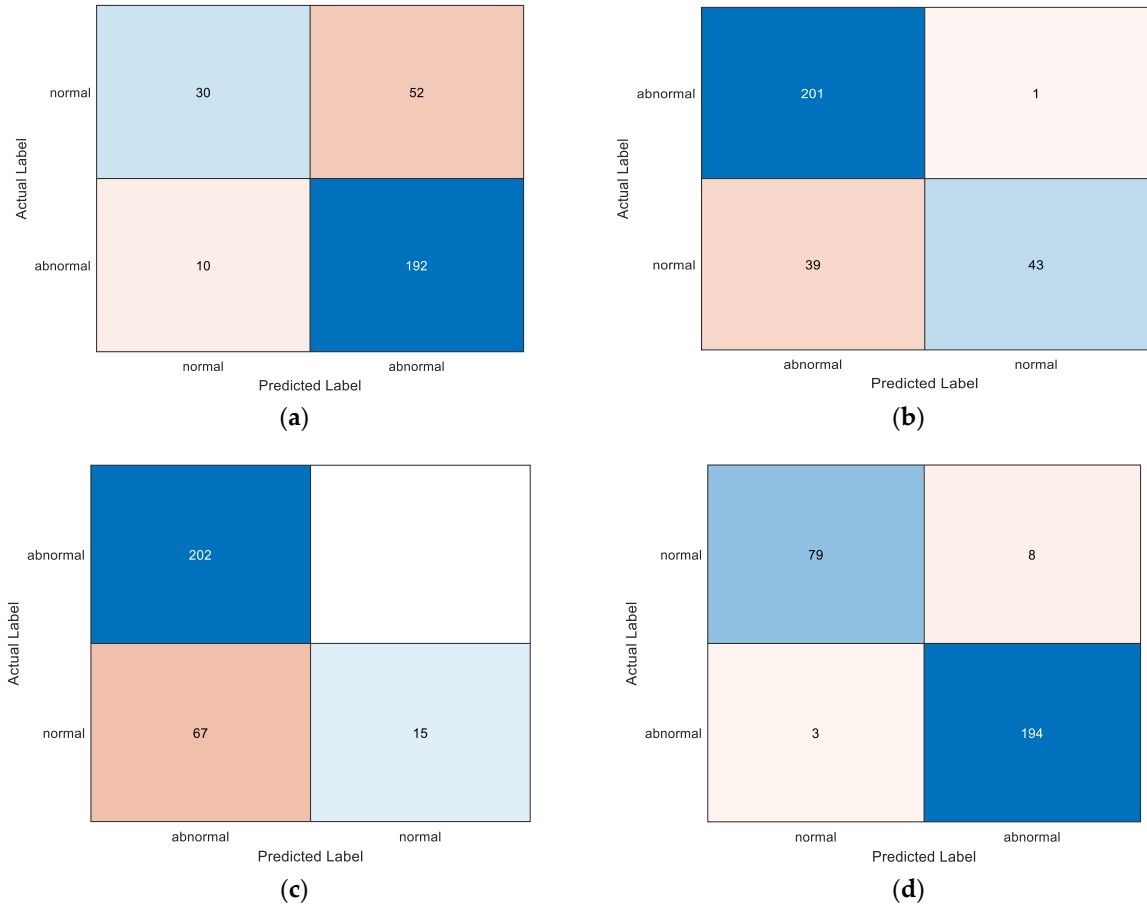

**Figure 12.** Confusion matrix obtained by different models: (**a**) ECG-PCG-BiLSTM; (**b**) ECG-GoogLeNet; (**c**) PCG-GoogLeNet; (**d**) BiLSTM-GoogLeNet-DS.

In order to show more intuitively and clearly the superiority of the proposed algorithm, the BiLSTM-GoogLeNet-DS model is compared with the BiLSTM model and the GoogLeNet model, respectively. The comparison results are shown in Table 3. It can be seen from Table 3 that the classification method proposed in this paper, based on BiLSTM-GoogLeNet-DS, has the best classification effect. Although the classification method based on PCG-BiLSTM obtained 100% sensitivity, 100% precision, and a 100% F1 score, the specificity is only 0%. This means that the classification method cannot identify healthy samples. Although the classification method based on PCG-GoogLeNet obtained 100% sensitivity, the specificity is only 18.29%. This means that the classification effect of this method for health samples is also very unsatisfactory.

**Table 3.** Comparison of several algorithm models.

| Classification Mode | Accuracy | Sensitivity | Specificity | Precision | F1 Score |
|---|---|---|---|---|---|
| PCG-BiLSTM | 71.13% | 100% | 0% | 100% | 100% |
| ECG-BiLSTM | 75% | 91.58% | 34.15% | 77.41% | 83.9% |
| PCG-ECG-BiLSTM | 78.17% | 95.05% | 36.59% | 78.69% | 86.1% |
| PCG-GoogLeNet | 76.41% | 100% | 18.29% | 75.09% | 85.78% |
| ECG-GoogLeNet | 85.92% | 99.51% | 52.44% | 83.75% | 90.95% |
| BiLSTM-GoogLeNet-DS | 96.13% | 98.48% | 90.8% | 96.04% | 97.24% |

In order to express the comparison results in the table more clearly, this paper uses the following histograms for graphical displays. It can be clearly seen from Figure 13 that the classification accuracy of the algorithm proposed in this paper (based on the BiLSTM-GoogLeNet-DS model) is higher than that of other classification models. In Figure 14, the

sensitivity, specificity, precision, and F1 score are compared. The best classification results are obtained by using the BiLSTM-GoogLeNet-DS model. The classification accuracy, sensitivity, specificity, precision, and F1 score are 96.13%, 98.48%, 90.8%, 96.04%, and 97.24%, respectively. From a comprehensive point of view, the algorithm proposed is better than other algorithm models.

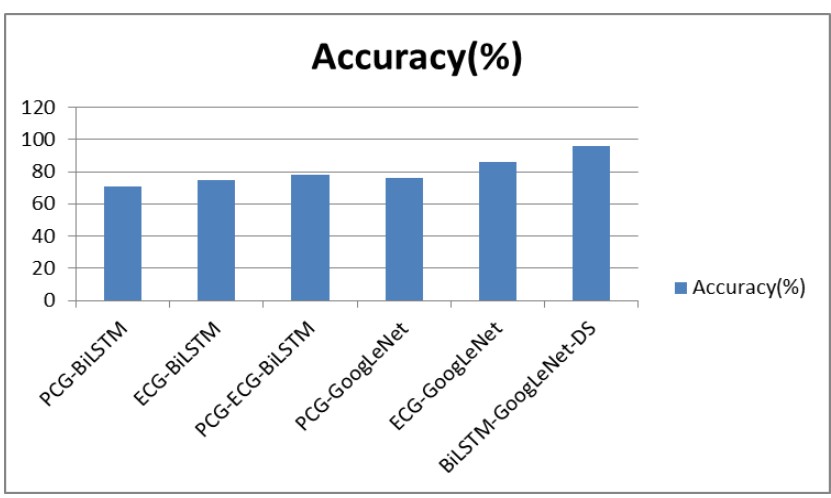

**Figure 13.** The comparison of accuracy.

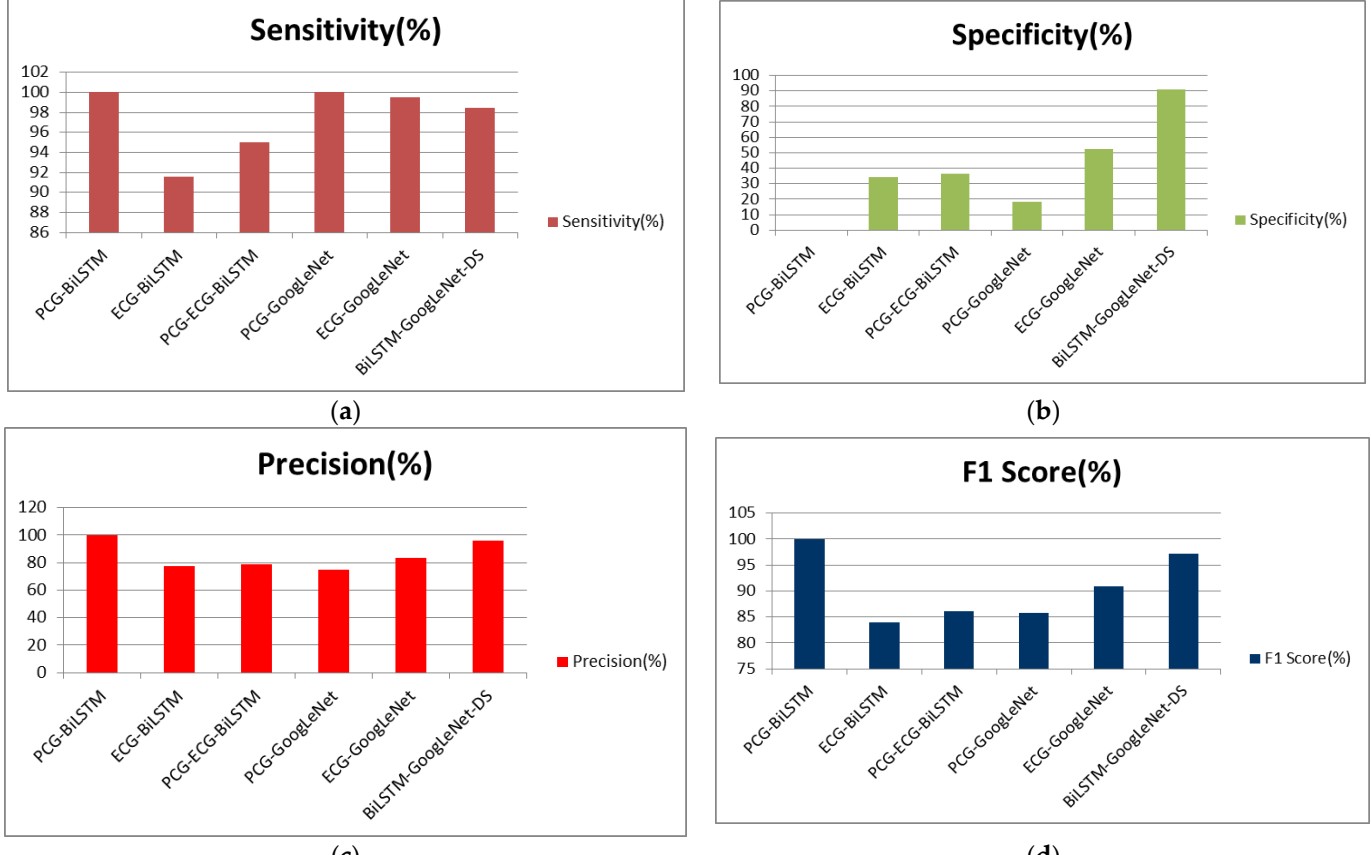

**Figure 14.** The comparison of sensitivity, specificity, precision, F1 score: (**a**) the comparison of sensitivity; (**b**) the comparison of specificity; (**c**) the comparison of precision; (**d**) the comparison of F1 score.

This paper also compares the methods from other literature studies. The comparison mainly focuses on the methods based on deep learning and the classification signals. Moreover, the current research studies on ECG and PCG signals mainly focus on the database provided in the PhysioNet/CINC challenge 2016 heart sound database, so the comparative classification effect is more convincing. The comparison results are shown in Table 4. The feature form is very similar, mainly focus on time-domain, frequency-domain, and time-frequency domain. The classification process is performed by 1D or 2D convolutional layers, and so on. The researched signal is single-modal or ECG and PCG signals. From the results, the classification effect of the method proposed in this paper is better than the methods in the literature, which shows that the algorithm proposed in this paper has certain advantages. The results also show that the classification method based on ECG and PCG signals has a stronger discriminative ability than those based on single-modal signals. Among the methods based on ECG and PCG signals, the classification effect of the method proposed in this paper is the best. Although the classification effect of the proposed method is close to that of Li Han's [5] method, it also shows that the method proposed in this paper has slight advantages.

**Table 4.** Comparison between this algorithm and other algorithms in the literature.

| Literature | Classification Method | Feature Form | Results |
|---|---|---|---|
| Tschannen [29] | 1-D CNN | Time-frequency domain features PCG signal | Acc = 87.0% Sen = 90.8% Spe = 83.2% |
| Rubin [30] | 2-D CNN | MFCC features PCG signal | Acc = 84.8% Sen = 76.5% Spe = 93.1% |
| Noman [31] | 2-D CNN | MFCC features PCG signal | Acc = 88.8% Sen = 86.1% Spe = 91.6% |
| Chakir Fatima [7] | Support vector machine | PCG signal ECG signal | Acc = 92.5% Sen = 92.31% Spe = 92.86% |
| Li Han [5] | Dual-Input Neural Network | PCG signal ECG signal | Acc = 95.6% Sen = 98.5% Spe = 89.2% |
| This paper | BiLSTM-GoogLeNet-DS | PCG signal ECG signal | Acc = 96.13% Sen = 98.48% Spe = 90.8% |

## 6. Conclusions

In this paper, an ECG–PCG signal classification method based on BiLSTM-GoogLeNet-DS is proposed. The method filters the obtained ECG and PCG signals, and uses the BiLSTM network to fuse and classify the two cardiac function signals. Then, the time-frequency processing of the filtered signals is performed. The time-frequency diagrams of signals are input into the GoogLeNet network for classification. Combining the above classification results provides a discriminative decision using the improved D-S theory to obtain the final classification results.

The following conclusions can be drawn from the experiments:

(1) The proposed ECG and PCG signal classification method based on the BiLSTM-GoogLeNet-DS model is superior to the classification methods of the BiLSTM model and the GoogLeNet model. The classification method of ECG and PCG signals is better than the classification methods of single-modal signals.

(2) This paper uses the deep learning-oriented hybrid fusion method. After the feature layer fusion of signals, the improved D-S theory is used to fuse the decision layer. The better classification results are obtained.

(3) The BiLSTM model is used for feature layer fusion, and the obtained classification results are better than those of single-modal signals. This method can extract the features of long-distance-dependent information according to the BiLSTM network, which can improve the training time and learn the features better. It can produce higher classification accuracy for long-span time series data.

(4) The classification effect obtained by using the GoogLeNet model to classify cardiac function signals is better than the classification method based on the BiLSTM model, which highlights the advantages of GoogLeNet in classification and recognition.

(5) The classification results of GoogLeNet and BiLSTM are fused by the improved D-S theory for the decision-making layer, which mainly calculates the BPA function for the classification result of each channel. Using the local recognition credibility information generated by the confusion matrix to weigh the BPA function can improve the final classification result.

(6) The best classification results are obtained by applying the ECG and PCG signal classification method based on BiLSTM-GoogLeNet-DS with 96.13% classification accuracy, 98.48% sensitivity, 90.8% specificity, 96.04% precision, and a 97.24% F1 score.

It can be concluded from the experimental results that the information fusion of ECG and PCG signals improves the performance indicators of classification and improves the classification effect. How to fuse more modal cardiac function signals to be used for disease diagnosis and treatment will be the next research direction.

**Author Contributions:** Conceptualization, J.L. and L.K.; methodology, J.L.; software, J.L.; valida-tion, Q.D. and X.D.; formal analysis, X.C.; investigation, J.L.; resources, Q.D.; data curation, J.L.; writing—original draft preparation, J.L.; writing—review and editing, L.K.; visualization, X.D.; supervision, X.C.; project administration, J.L.; funding acquisition, L.K. and J.L. All authors have read and agreed to the published version of the manuscript.

**Funding:** This research was funded by the National Nature Science Foundation of China, grant number 52077143; Department of Education of Liaoning Province, grant number LZGD2020002, LJKZ0131; The Fundamental Research Funds in Heilongjiang Provincial Universities of China, grant number 135409424 and Heilongjiang Science Foundation Project, grant number ZD2019F004.

**Institutional Review Board Statement:** Not applicable.

**Informed Consent Statement:** Not applicable.

**Data Availability Statement:** The database is supported by the PhysioNet/CinC Challenge 2016.

**Acknowledgments:** We acknowledge the database support from the PhysioNet/CinC Challenge 2016.

**Conflicts of Interest:** The authors declare no conflict of interest.

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
