# Peer review of "Research on the Classification of ECG and PCG Signals Based on BiLSTM-GoogLeNet-DS"

_applsci, doi:10.3390/app122211762_

Round 1
Reviewer 1 Report
This paper combines BiLSTM, GooLeNet, and DS as a method to classify PCG and ECG signals resulting in an accuracy of 96.13%. The authors descript the details of BiLSTM and GooLeNet with equations and structural diagrams, which help the reader fully understand how these algorithms work. Besides that, the author compares the various algorithms in the article and also compares them with those of other articles. The results were presented in figures and tables which could observe the differences. In my opinion, this paper is acceptable to be published after revision. Here are my comments,
1. Grammar and typos: the author should pay more attention to the English writing skills, for example, on Page 3 Line 105 The classification decision is made by using the fusion strategy of improved D-S theory. The world using should be removed.; On Page 1 Line 42, When seeing a doctor is a quite confused expression; While on Page 2 Line 89, the author refers to D-S evidence theory, the authors need to re-examine the use of terminology.
2. Academic discipline: The author should use a formal email address instead of an informal email address. The author and the email addresses cannot be matched by the present information, which causes uncertain identity. The authors should use the email address with edu, ac,or other addresses from confirmed organizations.
3. Solid supplementary information: Since the authors’ project is algorithm based, it is suitable to provide the codes as supplementary information to support the project. There are multiple platforms to submit codes such as GitHub.
4. Lack of solid reference:
The authors should pay more attention to certain important reference papers. These are some examples of a lack of references.
a. On Page 1 Line 39, the author refers ECG examination is a common method in modern cardiovascular diagnosis without any reference
b. On Page 2 Line 50, the author refers In the previous related research without any reference
5. Misleading Figure: In Figure1 following the arrow, why the PCG signal perform the same pre-processing as ECG signal? If there is any difference between the pre-processing, the diagram should be different since the author refers to different filtering on Page 7 Line 242-245. Also, in Figure 8 the author refers the compassion before and after filtering for ECG signals, where is the result for compassion before and after filtering PCG signals?
6. Equation 8: the explanation of the tanh function is probably wrong. Please recheck.
7. Result analysis: some of the results reach extremely high values, for example, the PCG-GoogLeNet has 100% sensitivity, the authors should try to explain these values.
8. Comparison with other projects: The result compared with Li Han’s classification method of Dual-Input Neural Network in Table 4 is very close to BiLSTM-GoogLeNet-DS. Are there any other criteria to be compared to show the advanced ability of BiLSTM-GoogLeNet-DS? For example, the training time.

Author Response
Point 1: Grammar and typos: the author should pay more attention to the English writing skills, for example, on Page 3 Line 105 The classification decision is made by using the fusion strategy of improved D-S theory. The world using should be removed.; On Page 1 Line 42, When seeing a doctor is a quite confused expression; While on Page 2 Line 89, the author refers to D-S evidence theory, the authors need to re-examine the use of terminology.
Response 1: The English writing has been checked. The word “using” has been deleted on Page 3 Line 110. The expression”seeing a doctor” has been corrected on Page 1 Line 43, 44. The terminology D-S evidence theory has been changed to D-S theory in this revised manuscript on Page 2 Line 92, Page 16 Line 488.
Point 2: Academic discipline: The author should use a formal email address instead of an informal email address. The author and the email addresses cannot be matched by the present information, which causes uncertain identity. The authors should use the email address with edu, ac,or other addresses from confirmed organizations.
Response 2: The email address has been changed on Page 1 Line 6 in this revised manuscript.
Point 3: Solid supplementary information: Since the authors’ project is algorithm based, it is suitable to provide the codes as supplementary information to support the project. There are multiple platforms to submit codes such as GitHub.
Response 3: We have provided the codes as supplementary information to support the project.
Point 4: Lack of solid reference:
The authors should pay more attention to certain important reference papers. These are some examples of a lack of references.
- On Page 1 Line 39, the author refers ECG examination is a common method in modern cardiovascular diagnosis without any reference
- On Page 2 Line 50, the author refers In the previous related research without any reference
Response 4: The reference has been added on Page 1 Line 42 in the revised manuscript. The inaccurate content description has been corrected on Page 2 Line 52. We think that the reference is not needed here.
Point 5: Misleading Figure: In Figure1 following the arrow, why the PCG signal perform the same pre-processing as ECG signal? If there is any difference between the pre-processing, the diagram should be different since the author refers to different filtering on Page 7 Line 242-245. Also, in Figure 8 the author refers the compassion before and after filtering for ECG signals, where is the result for compassion before and after filtering PCG signals?
Response 5: The pre-processing in these two places refers to the operation that both ECG and PCG need to undergo pre-processing, which is the pre-processing in a broad sense, and does not refer to the same pre-processing method. Because the noise of PCG signals is little, the effect before and after filtering is not very obvious. The figure about PCG signals is not provided here.
Point 6: Equation 8: the explanation of the tanh function is probably wrong. Please recheck.
Response 6: The Equation 8 has been modified in the revised manuscript on Page 5 Line 156.
Point 7: Result analysis: some of the results reach extremely high values, for example, the PCG-GoogLeNet has 100% sensitivity, the authors should try to explain these values.
Response 7: These evaluation parameters are calculated according to Equations (19)-(23). The 100% sensitivity means that the positive signals can be classified correctly by the models PCG-BiLSTM and PCG-GoogLeNet. However, because the negative signals cannot be classified, the 0% specificity is obtained. The 100% precision and 100% F1 Score are obtained according to equations and experimental results in the model PCG-BiLSTM.
Point 8: Comparison with other projects: The result compared with Li Han’s classification method of Dual-Input Neural Network in Table 4 is very close to BiLSTM-GoogLeNet-DS. Are there any other criteria to be compared to show the advanced ability of BiLSTM-GoogLeNet-DS? For example, the training time.
Response 8: There is not other criteria for comparison in the Li Han’s article. So we are sorry that the other comparison cannot be provided.

Reviewer 2 Report
1- In the introduction, the structure of the paper and the following sections should be mentioned.
2- If the number of training data is reduced, how will the results change?
3- If the training of the network is repeated 10 times and randomly, what will be the average and standard deviation of the results?
4- In addition to the criteria assumed in the paper, the evaluation criteria should be calculated according to the following paper:
Liu C, Springer D, Li Q, Moody B, Juan RA, Chorro FJ, Castells F, Roig JM, Silva I, Johnson AE, Syed Z, Schmidt SE, Papadaniil CD, Hadjileontiadis L, Naseri H, Moukadem A, Dieterlen A, Brandt C, Tang H, Samieinasab M, Samieinasab MR, Sameni R, Mark RG, Clifford GD. An open access database for the evaluation of heart sound algorithms. Physiological Measurement 2016;37(9).
Author Response
Point 1: In the introduction, the structure of the paper and the following sections should be mentioned.

Response 1: The structure of the paper and the following sections have been added on Page 2-3 Line 95-103.
Point 2: If the number of training data is reduced, how will the results change?
Response 2: If the number of training data is reduced, the classification accuracy will improve.
Point 3: If the training of the network is repeated 10 times and randomly, what will be the average and standard deviation of the results?
Response 3: If the training of the network is repeated 10 times and randomly, the average and standard deviation of the classification accuracy is respectively 96.13% and 0.
Point 4: In addition to the criteria assumed in the paper, the evaluation criteria should be calculated according to the following paper:
Liu C, Springer D, Li Q, Moody B, Juan RA, Chorro FJ, Castells F, Roig JM, Silva I, Johnson AE, Syed Z, Schmidt SE, Papadaniil CD, Hadjileontiadis L, Naseri H, Moukadem A, Dieterlen A, Brandt C, Tang H, Samieinasab M, Samieinasab MR, Sameni R, Mark RG, Clifford GD. An open access database for the evaluation of heart sound algorithms. Physiological Measurement 2016;37(9).
Response 4: Because this paper mainly compares the experimental data of multi-modal signals with the experimental data of single-modal signals using the deep learning classification method. The selection of evaluation criteria about the compared literature is consistent, so the evaluation criteria of the mentioned literature is not used.
Reviewer 3 Report
The abstract needs quantification. Since PCG and ECG are operating in the different domain how it is possible to have a fusion of these signals. PCG has distinct state based signals and hence the syn of ECG would not produce good accuracy. The binary classification is a crude one for an CNN based system. Preprocessing steps has to be analyzed in a proper manner. There is no novelty in the paper. Jaccard Index and MCC has to be calculated. Table 3 and 4 needs more explanation.
Author Response
Point: The abstract needs quantification. Since PCG and ECG are operating in the different domain how it is possible to have a fusion of these signals. PCG has distinct state based signals and hence the syn of ECG would not produce good accuracy. The binary classification is a crude one for an CNN based system. Preprocessing steps has to be analyzed in a proper manner. There is no novelty in the paper. Jaccard Index and MCC has to be calculated. Table 3 and 4 needs more explanation.
Response:
The information fusion mentioned in this paper is to fuse the processed signals at the decision-making layer instead of directly fusing the PCG and ECG signals.This fusion is independent of the signal state and operational domain.
The preprocessing is mainly to filter the signals. Since the preprocessing is for the data in the database, the noise is not complicate, the code is open and it is not the focus of this paper. Therefore, this part of the content has not been emphatically analyzed.
In order to highlight the advantages of the algorithm proposed in this paper, the algorithm proposed in this paper is compared with the algorithms of other literature. These two parameters Jaccard Index and MCC are not calculated for these literature, so we think this paper does not need to add these two evaluation parameters.
The explanation has been added about Table 3 on Page 14 Line 448-455 and Table 4 on Page 16 Line 487-490 in the revised manuscript.

Round 2
Reviewer 1 Report
The problems I refered in my previous report has been well solved in the author's reply. Here I don't have further comments.
Author Response
Point:The problems I refered in my previous report has been well solved in the author's reply. Here I don't have further comments.
Response:Thanks for the reviewers' approval.
Reviewer 3 Report
Point to point reply is required with technical proof. We are unable to accept the reply given by the authors for our queries.
